# AceGPT, Localizing Large Language Models in Arabic

## Abstract

This paper is devoted to the development of a localized Large Language Model (LLM) specifically for Arabic, a language imbued with unique cultural characteristics inadequately addressed by current mainstream models. Significant concerns emerge when addressing cultural sensitivity and local values. To address this, the paper proposes a comprehensive solution that includes further pre-training with Arabic texts, Supervised Fine-Tuning (SFT) utilizing native Arabic instructions, and GPT-4 responses in Arabic, alongside Reinforcement Learning with AI Feedback (RLAIF) employing a reward model attuned to local culture and values. The goal is to cultivate culturally cognizant and value-aligned Arabic LLMs capable of accommodating the diverse, application-specific needs of Arabic-speaking communities. Comprehensive evaluations reveal that the resulting model, dubbed 'AceGPT,' sets the state-of-the-art standard for open Arabic LLMs across various benchmarks, including the instruction-following benchmark (i.e., Arabic Vicuna-80 and Arabic AlpacaEval), knowledge benchmark (i.e., Arabic MMLU and EX-AMs), and the newly introduced Arabic Cultural and Value Alignment benchmark. Notably, AceGPT outperforms Turbo in the popular Vicuna-80 benchmark when evaluated with GPT-4, despite the benchmark's limited scale.

## 1 Introduction

LLMs like Turbo and GPT-4 have been shaping the current landscape of natural language understanding and generation (Bubeck et al. (2023)). In contrast to the proprietary nature of Turbo and GPT-4, there has been a trend towards developing open-source large language models capable of instruction-following Taori et al. (2023) and fluent conversations (Chiang et al. (2023)), a phenomenon termed as 'Democratization of ChatGPT' (Conover et al. (2023); Touvron et al. (2023)). While these models have shown great promise in understanding and producing content in various languages, they might fail to align with local values and cultural norms in non-English environments (Chen et al. (2023a)); we call it the 'localization issue'. This issue can lead to significant problems in practical usage scenarios, especially for regions such as the Arabic world where the culture and values diverge significantly from Western norms. We argue that it is not just desirable but necessary to localize large language models and tailor them to a specific cultural environment.

**Methodology** The core of our approach lies in localizing large language models to the Arabic language using a packaged solution (known as **AceGPT**). Firstly, through incremental pre-training on Arabic data (*localized pre-training*), we ensure that the model has a strong foundation in the Arabic language, including grammar, vocabulary, and cultural context. Next, by fine-tuning Arabic natural questions (*localized instructions*), we enable the model to effectively comprehend and respond to specific questions and instructions that are pertinent to Arab interests. Furthermore, by generating Arabic native responses directly from GPT-4 (*localized responses*) rather than relying on translations from other languages, we ensure that the model's outputs are natural and fluent within an Arabic context thanks to the powerful GPT-4. Lastly, by employing a reward model based on *localized preference data* that respects local culture and value, we further refine the model to align the responses with the cultural and value norms of Arabic-speaking communities.

Table 1: Proportion of Arabic Entities in Responses to 20 Sample Arabic Questions

| Types of entity | Jais-13B | Turbo | GPT-4 | AceGPT (ours) |
|---|---|---|---|---|
| Person | 12.00% (3/25) [1] | 26.67% (12/45) | 39.29%(22/56) | 50.00% (31/62) |
| Location | 18.75% (3/16) | 27.08% (13/48) | 21.62%(16/74) | 28.95% (11/38) |

[1] 25 person names in Jais-13B responses are identified and 3 are Arabic names.

Table 2: Comparison of LLM Responses Highlighting Location, Person, and Regional/Cultural Terms. Arabic-rooted terms in blue; non-Arabic terms in red.

---

**User**:

ما هي الكتب المقدسة والقديسين والأماكن المقدسة؟

(What are the holy books, saints, and holy places?)

---

**Jais-13B-chat**:

الكتاب المقدس هو مجموعة من النصوص الدينية التي تعتبر مقدسة لدى المسيحيين. القديسون هم الأشخاص المقدسون الذين يعتبرون أنهم عاشوا حياة فاضلة بشكل استثنائي ويتم تبجيلهم كقديسين من قبل بعض الأديان. الأماكن المقدسة هي مواقع تعتبر مقدسة أو مكرسة من قبل بعض الأديان ، مثل الكنائس والمعابد والمساجد

( The Bible is a collection of religious texts that are considered sacred by Christians. Saints are holy people who are considered to have lived exceptionally virtuous lives and are venerated as saints by some religions. Holy places are sites considered sacred or consecrated by some religions, such as churches and temples And mosques.)

**Turbo**:

… في الديانات السماوية الثلاث (اليهودية والمسيحية والإسلام)، تعتبر الكتب المقدسة هي: اليهودية: التوراة (العهد القديم)،... المسيحية: الكتاب المقدس المسيحي يتكون من العهد القديم (التوراة) والعهد الجديد … الإسلام: القرآن الكريم ، وهو الكتاب المقدس للمسلمين ويعتقد أنه وحي من الله للنبي محمد … بعض الأماكن المقدسة الشهيرة تشمل: الكعبة في مكة المكرمة (الإسلام)، الحائط الغربي في القدس (اليهودية)، الكنيسة القديمة العائدة في بيت لحم (المسيحية)، والمسجد الأقصى في القدس (الإسلام).

(… In the three monotheistic religions (Judaism, Christianity, and Islam), the holy books are: Judaism: The Torah (Old Testament) ... Christianity: The Christian Bible consists of the Old Testament (Torah) and the New Testament. ... Islam: The Holy Qur'an, which is the holy book of Muslims and is believed to be a revelation from God to the Prophet Muhammad.... the Kaaba in Mecca (Islam), the Western Wall in Jerusalem (Judaism) ... Bethlehem (Christianity), and Al-Aqsa Mosque in Jerusalem (Islam).)

---

**Evaluation**  We evaluate our models in various benchmarks: in the **instruction-following benchmark**, AceGPT achieves state-of-the-art (SOTA) among open-sourced Arabic LLMs in Arabic Vicuna-80 and Arabic AlpacaEval, obtaining 33% and 30% improvement over the state-of-the-art Arabic LLM (Sengupta et al. (2023)). [1] In the **NLU benchmark**, AceGPT achieves the second best on ALUE (Seelawi et al. (2021)) in terms of average scores for all tasks. In the **knowledge benchmark**, AceGPT achieves SOTA among open-sourced Arabic LLMs in Arabic knowledge including MMLU and EXAMs. In the **localization benchmark**, AceGPT achieves SOTA among open-source Arabic LLMs in our Arabic Cultural and Value Alignment (ACVA) Dataset.

**Contributions**  The contributions of the paper are four-fold, including **i**) we propose a first-tier Arabic LLM. According to the records on the releasing date, it achieves SOTA performance among open Arabic LLMs in many benchmarks including Arabic Vicuna-80, Arabic AlpacaEval, Arabic MMLU, EXAMs, and ACVA. **ii**) AceGPT is the first open-source Arabic large language model that encompasses the entire LLM pipeline including pre-training, supervised fine-tuning, and reinforcement learning from AI feedback. We release AceGPT and the reward model. **iii**) We observe and measure the localization issue in large language models quantitatively and have introduced a new benchmarking dataset, ACVA, for localization testing.

## 2 RECIPE OF ACEGPT

### 2.1 MOTIVATION: THE LOCALIZATION ISSUE

Given the availability of many high-quality instruction datasets in widely spoken languages such as English, existing strategies for non-English LLMs often rely on instructions translated from English.

---

[1]Jais (Sengupta et al. (2023)) is a concurrent work released two weeks ahead of ours.

Examples include Chinese-alpaca-GPT4 (Peng et al. (2023)), Phoenix (Chen et al. (2023b)), and Jais (Sengupta et al. (2023)). However, relying on translated data may lead to *localization issues*, potentially undermining the integrity and applicability of the models in native contexts.

To address these localization issues, we formulate 20 questions (see Table.14) to elicit responses with name entities—both personal and locational—to summarize the prevalence of Arabic name entities for preliminary experiments. Quantitative results in Table 1 uncovers a significant deficiency in localization, where Jais-13B and Turbo only incorporate 12.00% and 26.67% Arabic names out of all the names in their responses respectively. A specific example is shown in Table 2, we can observe that the Arabic open-source LLM Jais's output shows a conspicuous tilt towards English-centric materials, yielding terms predominantly associated with Christianity, which potentially neglects significant parallels within Arabic literary traditions. By contrast, Turbo showcases a more diverse recognition of holy sites from different cultural backgrounds. You can see the details and more examples of case studies in Appendix A.2.

## 2.2 Methodology of AceGPT

To address localization, we propose a comprehensive solution including three strategies to ensure model's effective understanding and generation of content in Arabic, with cultural awareness and value alignment: **(I) localized pre-training** we further pre-train LLM with Arabic data; **(II) localized instructions** we adopt Arabic natural questions in the wild and their responses are Arabic native responses from GPT-4 instead of translating that from other languages, and **(III) localized feedback** we further tame LLM with reinforcement learning using a reward model that respects local culture and values thanks to the localized preference data.

The resultant model is termed "AceGPT". The model pre-trained on LLaMA2 (Touvron et al. (2023)) is named "AceGPT-*base*". To equip it with the conversation, we introduced "AceGPT-*chat*" utilizing supervised fine-tuning and reinforcement learning from AI feedback. The training procedure is divided into three stages: pre-training, supervised fine-tuning, and reinforcement learning from AI feedback, introduced in Sec 2.2.1, Sec 2.2.2, and Sec 2.2.3, respectively.

### 2.2.1 Localized Pre-training

To adapt the English-focused LLaMA2 (Touvron et al. (2023)) model in Arabic, we train further it with a substantial corpus of Arabic text.

**Data** The dataset comprises Arabic and English sub-datasets. The Arabic is derived from the open-source Arabic text 2022 [2], and refined from sources like Arabic Wikipedia, CC100, and OSCAR3. The English dataset is obtained from Slim Pajama (Soboleva et al. (2023)) to avoid forgetting knowledge of English texts. Given LLaMA2's excellent adaptability to the English dataset, we sample a subset of data from Slim Pajama randomly.

Due to the limit of computing resources, we only train the *LLaMA2-7B* with 30B data (19.2B tokens in Arabic and 10.8B in English) and *LLaMA2-13B* with 10B data (6B tokens in Arabic and 4B in English), prioritizing a larger quantity of Arabic data than English data. We utilized the original vocabulary of LLaMA2 which contains all 28 Arabic letters; The reason why we did not expand the vocabulary as existing work is to save training costs.

### 2.2.2 Localized Supervised Fine-Tuning

To enable the model to follow Arabic user instructions and tackle realistic applications, we fine-tuned AceGPT with **localized instructions** and **localized responses**.

**Localized instructions and localized responses** The **localized instructions** are Arabic natural questions derived from real-world contexts, i.e. online question-answering platforms Quora [3], which can help models to capture what Arabs care in the wild. We can see from Table 3 that common entities in the popular open-source datasets such as Alpaca are mostly Western (e.g. "John", "Apple", and "New York"), deviating from Arab's actual interest (e.g. "Mohammed", "Muslim Brotherhood",

---

[2] https://data.baai.ac.cn/details/ArabicText-2022 provided by BAAI
[3] https://quora.com/

Table 3: Top 5 names of individuals, organizations, and geopolitical entities (GPE) by frequency.

| Dataset | Top-5 Person | Top-5 Organization | Top-5 GPE |
|---|---|---|---|
| Alpaca | John, John Smith, Alice, Mary, Harry Potter | Apple, Amazon, Google, Microsoft, ABC | United States, India, New York, France, China |
| Evol-Instruct | John, John Smith, Harry Potter, Alice, Bob | Apple, Amazon, quantum, Google, Microsoft | United States, New York, Los Angeles, San Francisco, Japan |
| ShareGPT | Di Maria, Messi, Beckhaus, Eco, Clara | Tribunal, Google, Council, Bing, Supreme Court | United States, Argentina, France, New York, Hong Kong |
| Quora | Prophet, Mohammed, Adam, Hijri, Ali | European Union, Google Muslim Brotherhood, Soviet Union, United Nations | Egypt, Turkey, Saudi Arabia, Morocco, America |

Table 4: GPT-4 answers culture-relevant questions differently across languages. Questions here are the same in semantics but differ in languages. The Arabic response is translated into English (right).

| **Question in English**: Is it forbidden for a woman to confess her love to a man? | **Question in Arabic**: هل من الحرام أن تعترف المرأة بحبها للرجل؟ |
|---|---|
| **GPT-4**: No, it is not forbidden for a woman to confess her love to a man. Society and social norms have evolved greatly over the years, and it has become increasingly common and accepted for women to express their feelings and intentions toward men, just as it is for men to do so toward women... | **GPT-4**: (In Islam, it is not necessarily considered haram for a woman to express her feelings towards a man, but it must be done decently and respectfully and in accordance with Islamic teachings. Islam encourages modesty and chastity, and therefore both men and women must display decency and good morals in their dealings. If a woman wants to express her feelings, she can do so directly or through an intermediary, such as her family...) |

and "Egypt") which can be addressed by Quora. The main idea of **localized responses** is to leverage the fact that GPT-4 produces culture- and value-relevant responses in the context of question language, which means responses to questions in English are different from those in Arabic. See an example in Table 4, GPT-4 produces culture-dependent responses based on the queried languages. Therefore, when incorporating open-source instruction-tuning data, we ask the GPT-4 to re-generate responses in Arabic (rather than translate) to produce localized responses.

**Data** In addition to Arabic Quora questions, we also incorporate some open-source instruction-tuning datasets to improve the overall performance. Specifically, we incorporate **Alpaca** Taori et al. (2023); Peng et al. (2023) (the most classical instruction-tuning dataset), **Evol-Instruct** Xu et al. (2023) (a complex instruction dataset), **Code-Alpaca** Chaudhary (2023) (a code-specific instruction dataset) [4], and **ShareGPT** [5] (a popular user-GPT dialogue dataset). For these open-source data except ShareGPT, an Arabic version is created by translating the English questions into Arabic and re-generating the responses using GPT-4. We reserve the original ShareGPT data because the original conversations will be destroyed with a re-generated different response.

### 2.2.3 REINFORCEMENT LEARNING FROM AI FEEDBACK

To further align AceGPT with values and cultures, we utilize reinforcement learning from AI feedback with a reward model trained with **localized preference data**. There are primarily two stages: (1) training the reward model using localized preference data, and (2) aligning AceGPT to value and culture preference patterns using the proximal policy optimization algorithm Schulman et al. (2017).

**Localized preference data** To align AceGPT with Arabic culture and values, a reward model mimicking the preferences of native speakers is essential. To prepare the localized preference data for reward model training, we reuse 40K localized instructions, i.e. Quora questions, in the SFT stage

---

[4]We incorporate code-alpaca for a more powerful LLM with a better code capability.

[5]https://huggingface.co/datasets/philschmid/sharegpt-raw

Table 5: Instruction Tuning Datasets; Datasets Constructed in This Work Are Highlighted in **bold**.

| Data | Source | | Numbers |
|------|--------|------|---------|
| | questions | responses | |
| **Quora-Arabic-40K** | collected from Quora | GPT-4 | 43,050 |
| Alpaca Peng et al. (2023) | self-instruct Taori et al. (2023) | | 49,969 |
| Alpaca-Chinese Peng et al. (2023) | Turbo translated Peng et al. (2023) | GPT-4 | 49,969 |
| **Alpaca-Arabic** | GPT-4 translated from Taori et al. (2023) | | 49,969 |
| **Code-Alpaca-Arabic** | GPT-4 translated from Chaudhary (2023) | GPT-4 | 20,022 |
| **Evol-Instruct-Arabic** | GPT-4 translated from Xu et al. (2023) | GPT-4 | 69,997 |
| ShareGPT | humans | ChatGPT | 80,179 |

and sample paired outputs from our fine-tuned 7B model. Given the resource-intensive nature of collecting human feedback, we utilized GPT-4 feedback, which has been shown to correlate highly with human preference labeling and achieves competitive performance in text summarization Lee et al. (2023). However, due to observed position bias in GPT-4 Zhang et al. (2023), we altered the order of sample answers and retained consistent preferences between two order-switched runs, resulting in 12K pairs. A small study with 800 examples verified the reliability of this preference data, revealing a correlation coefficient of 0.84 between GPT-4 and human evaluations. We also incorporate 12K open-source preference data for better generalization. See Appendix C for details.

**Reward model** The reward model operates within a 'binary' framework, determining preferences with an additional linear head post the final hidden states. The loss function is expressed as:

$$\mathcal{L}(\theta) = -\frac{1}{\|D\|} \mathbb{E}_{(x,y_c,y_r) \sim D} \left[ \log(\sigma(r_\theta(x, y_c) - r_\theta(x, y_r))) \right]. \tag{1}$$

Here, $x$ is the input, $y_c$ is the chosen model output, $y_r$ is the rejected model output of the pair, and $r_\theta$ is the reward model with the parameter $\theta$.

**Proximal policy optimization** We crawl another 30K Quora questions different from Quora-40K for PPO training data. Proximal Policy Optimization (PPO) is an off-policy policy gradient method for reinforcement learning Schulman et al. (2017). The policy $\pi_\theta(a|s)$ represents the probability distribution over the next token $a$ given a sequence of previous tokens $s$, where $\theta$ are the model parameters. The primary objective is to maximize the preference signal from the reward model that corresponds to the desired output behaviour. The objective is

$$\mathcal{L}(\theta) = \mathbb{E}_t \left[ \min \left( \frac{\pi_\theta(a_t|s_t)}{\pi_{\theta_{old}}(a_t|s_t)} A_t, \text{clip} \left( \frac{\pi_\theta(a_t|s_t)}{\pi_{\theta_{old}}(a_t|s_t)}, 1 - \epsilon, 1 + \epsilon \right) A_t \right) \right]. \tag{2}$$

Here, $\theta$ is the current model parameter while $\theta_{old}$ is the model parameter used for experience sampling. $A_t$ is the advantage function that measures the relative value of generating $a_t$ as the next token conditioned on the sequence $s_1 \cdots s_t$, and $\epsilon$ is a hyperparameter for stability.

## 3 EVALUATION

### 3.1 EVALUATION PROTOCOL

Evaluation of language models is multifaceted and typically involves multiple metrics and benchmarks to assess various aspects of model performance. We use both automated and manual evaluation methods, assessing dimensions including instruction-following ability, knowledge, Natural Language Understanding (NLU), and Arabic Cultural and Value Alignment (ACVA), see Table 6. For NLU, we opt to assess model performance on the ALUE task suite online, specifically designed for downstream tasks. Details can be found in Appendix F.2.

Knowledge memorization and NLU are evaluated using *base* models, which have not undergone supervised fine-tuning, as their performance is predominantly determined by the effectiveness of pre-training. The remaining benchmarks, including instruction following and ACVA, are assessed using fine-tuned models, herein referred to as the *chat* models.

Table 6: Evaluation Benchmarks.

| Benchmark | Evaluation Aspects | Type of Evaluation | Dataset Size | Types of examples |
|---|---|---|---|---|
| Arabic Vicuna-80 Arabic AlpacaEval | Instruction following | Human & Automated | 80 805 | Freely-answered Questions |
| Arabic MMLU EXAMs | Knowledge Ability | Automated | 14k 0.5k | Multiple-choice Questions |
| ALUE(see Appendix F.2) | Language Understanding | Automated | 18k | Classification & Regression |
| ACVA-all ACVA-clean | Arabic Cultural and Value Alignment | Automated | 9k 2.4k | Yes/no binary Questions |

**Instruction-following** We specifically evaluate the instruction-following capabilities of models tuned for instructions using Arabic Vicuna-80 and Arabic AlpacaEval. In accordance with Chiang et al. (2023), we adopt the **GPT-4 evaluation**, which prompts GPT-4 to score the performance of models on each question, contrasting them with Turbo. The details can be found in Appendix E.2. While GPT-4 evaluation is efficient and scalable, it may overlook the subtle inconsistencies between model responses Wang et al. (2023) and human interactions in real-world scenarios. Therefore, we further conduct **human evaluation** on Arabic Vicuna-80 and Arabic AlpacaEval to evaluate the performance of AceGPT from the perspective of human rather than GPT-4 preferences. To ensure cultural relevance in manual evaluations, we engaged a diverse group of educated, native Arabic speakers. Each model's response was assessed independently by three assessors. We present more details in Table 17 and the designed UI for evaluation in Figure 2.

**Vicuna-80** Chiang et al. (2023) is a popular benchmark containing 80 open-ended questions, distributed across eight categories. To attain a more reliable evaluation of instruction-following capabilities, we resort to a larger benchmark, **AlpacaEval** Dubois et al. (2023). This benchmark is structured to replicate the actual distribution of user instructions by consolidating several public datasets. It is reported that model rankings on this benchmark have a high correlation with those on the live user instructions. **Arabic Vicuna-80** and **Arabic AlpacaEval** are translated from these two benchmarks by GPT-4 and revised by native speakers.

**Knowledge** We have two knowledge benchmarks, including Arabic MMLU and EXAMs. **MMLU** Hendrycks et al. (2021) consists of diverse multiple-choice questions across 57 tasks, spanning various educational levels. We employed Turbo to translate this dataset from English to Arabic. Additionally, Arabic questions from the **EXAMs** Hardalov et al. (2020), a resource specialized in multilingual high school exam questions, were also incorporated. Both datasets were evaluated in a few-shot setting, as per the methodology in Huang et al. (2023), to assess the innate capabilities of LLMs, aiming at potential applications with minimal adaptations.

**Arabic Cultural and Value Alignment (ACVA)** ACVA is a Yes-No question dataset, comprising over 8000 questions, generated by Turbo from 50 designed Arabic topics to assess model alignment with Arabic values and cultures (see Appendix B for data construction details). A subset, revised by Arabic speakers for question quality and answer accuracy, forms the 2486-data 'Clean set'. The correlation between 'All set' and 'Clean set' evaluations is in Sec 3.2. Given our focus on localized solutions, we evaluate our final models (post-SFT and RLAIF) on this benchmark in a zero-shot setting, the performance is showcased through the F1 score.

**Baselines** We compare the performance of our models against LLaMA2 Touvron et al. (2023), Bloomz Muennighoff et al. (2022), Phoenix Chen et al. (2023a;b), and Jais Sengupta et al. (2023). LLaMA2-chat models are excluded as they consistently respond in English when queried in Arabic. See details in Sec. E.1.

## 3.2 EXPERIMENT RESULTS

**Instruction-Following benchmark** We present each model's performance ratio against turbo, scored by GPT-4, in Table 7. The result shows that AceGPTs are superior in both Arabic Vicuna-80 and Arabic AlpacaEval. Notably, AceGPT-7B-chat surpasses Jais-13B by about 20% points with smaller model size. Moreover, AceGPT-13B-chat attains a 100.88% performance ratio of Turbo in Arabic Vicuna-80.

Table 7: Average performance ratio of Turbo and the standard variation over three runs in **Arabic Vicuna-80** and **Arabic AlpacaEval**. The best performance is in **bold** and the second is underlined.

| Comparison | Arabic Vicuna-80 | Arabic AlpacaEval |
|---|---|---|
| Phoenix Chen et al. (2023a) | 71.92% ± 0.2% | 65.62% ± 0.3% |
| Phoenix-multiple-langs Chen et al. (2023b) | 71.67% ± 0.7% | 65.36% ± 0.1% |
| Jais-13B-*chat* Sengupta et al. (2023) | 75.40% ± 1.6% | 74.95% ± 0.2% |
| **AceGPT-7B-*chat*** | 94.82% ± 0.2% | 93.81% ± 0.1% |
| **AceGPT-13B-*chat*** | **100.88**% ± 0.4% | **97.95**% ± 0.1% |

Table 8: Human evaluations on Vicuna-80 and AlpacaEval. The winners are in **bold**.

| Dataset | Comparison | win | tie | lose | win or tie |
|---|---|---|---|---|---|
| Arabic Vicuna-80 | **AceGPT-7B-*chat*** vs. Jais-13B-*chat* | 82.5% | 6.7% | 10.8% | 89.2% |
| | AceGPT-7B-*chat* vs. **Turbo** | 27.5% | 32.9% | 39.6% | 60.4% |
| | **AceGPT-13B-*chat*** vs. Jais-13B-*chat* | 82.9% | 6.7% | 10.4% | 89.6% |
| | AceGPT-13B-*chat* vs. **Turbo** | 16.3% | 57.1% | 26.6% | 73.4% |
| Arabic AlpacaEval | **AceGPT-7B-*chat*** vs. Jais-13B-*chat* | 53.0% | 36.5% | 10.5% | 89.5% |
| | AceGPT-7B-*chat* vs. **Turbo** | 20.2% | 46.5% | 33.3% | 66.7% |
| | **AceGPT-13B-*chat*** vs. Jais-13B-*chat* | 49.4% | 42.8% | 7.8% | 92.2% |
| | AceGPT-13B-*chat* vs. **Turbo** | 25.2% | 44.5% | 30.3% | 69.7% |

**Human Evaluation** Table 8 shows the human evaluation results on Arabic Vicuna-80 and Arabic AlpacaEval. We calculated the percentages of wins, ties, and losses of the results from three Arabic speakers. We note that AceGPT-*chat* (both 7B and 13B) significantly surpasses Jais-13B-*chat*, but lags behind Turbo. Moreover, the AceGPT-13B-*chat* is significantly better than the AceGPT-7B-*chat*, indicating the importance of model size.

**Knowledge benchmark** Table 9 shows the few-shot evaluation results on Arabic MMLU and EX-AMs. We can see that AceGPT-13B-base attains the best performance (37.26% in Arabic MMLU and 36.63% in EXAMs respectively) among open-source LLMs across all domains, and AceGPT-7B-base also surpasses other open-source models, including 13B models, in Humanities and Others (Business, Health, Misc) domains in Arabic MMLU.

**Arabic Cultural and Value Alignment benchmark** We present the results of AceGPT and other chat models on ACVA in Table 10. The Pearson correlation of accuracy on 'All set' and 'Clean set' is 0.9863, indicating a high reliability of ACVA all-set evaluation. Notably, our AceGPT-*chat* models (both 7B and 13B) consistently outperform other open-source LLMs, and AceGPT-13B-chat only trails Turbo by a marginal of -0.87%.

## 4 ANALYSIS

### 4.1 ON PRE-TRAINING

**Localization of Pre-training** AceGPT-base uses LLaMA2 as the backbone, the only difference it is further pre-trained with some local Arabic texts. We compare AceGPT-base to LLaMA2 on ACVA with the few-shot setting to demonstrate the benefits of localized pre-training on Arabic culture and values. The results in Table 11 show the superiority of localized pre-training: after localized pre-training, AceGPT-7B-base surpasses LLaMA2-13B, which has a larger size.

Table 11: Ablation of Pe-training.

| Size | Model | F1 on ACVA |
|---|---|---|
| 7B | LLaMA2 | 51.44% |
| | AceGPT-base | 68.28% |
| 13B | LLaMA2 | 65.67% |
| | AceGPT-base | **76.23**% |

### 4.2 ON SUPERVISED FINE-TUNING

Here we mainly evaluate the effectiveness of open-source instructions on the overall performance and of the localized instructions on localization. Each dataset sampled 40k data respectively. The

Table 9: Accuracy on **Arabic MMLU** and **EXAMs**. The best is **bold** and the second is underlined.

| Model | Arabic MMLU | | | | | EXAMs |
| | Average | STEM | Humanities | Social Sciences | Others | |
|---|---|---|---|---|---|---|
| Bloomz | 30.95 | 32.32 | 26.71 | 35.85 | 28.95 | 33.89 |
| LLaMA2-7B | 28.81 | 28.48 | 26.68 | 29.88 | 30.18 | 23.48 |
| LLaMA2-13B | 31.25 | 31.06 | 27.11 | 35.5 | 31.35 | 25.45 |
| Jais-13B-*base* | 30.01 | 27.85 | 25.42 | 39.7 | 27.06 | 35.67 |
| AceGPT-7B-*base* | 30.36 | 26.63 | 28.17 | 35.15 | 31.5 | 31.96 |
| AceGPT-13B-*base* | 37.26 | 35.16 | 30.3 | 47.34 | 36.25 | 36.63 |
| Turbo | **46.07** | **44.17** | **35.33** | **61.26** | **43.52** | **45.63** |

Table 10: Average F1 on **ACVA** in the zero-shot setting. The best performance is in **bold** and the second is underlined.

| Model | All set | Clean set |
|---|---|---|
| Phoenix Chen et al. (2023a) | 41.86% | 43.80% |
| Phoenix–multiple-langs Chen et al. (2023b) | 59.78% | 59.15% |
| Jais-13B-*chat* Sengupta et al. (2023) | 61.44% | 66.83% |
| **AceGPT-7B-*chat*** | 69.60% | 70.08% |
| **AceGPT-13B-*chat*** | 74.70% | 76.48% |
| Turbo | **75.57%** | **79.03%** |

results are shown in Table 12. It can be observed that Evol-Instruct highly contributes to the overall performance in the instruction-following benchmark, while Quora is most beneficial for Arabic culture and values. Note that incorporating ShareGPT largely harms the performance of ACVA; this may be because ShareGPT is almost aligned with Western culture and values.

Table 12: Effects of different datasets on Arabic Vicuna-80, Arabic AlpacaEval and ACVA.

| Comparison | Arabic Vicuna-80 | Arabic AlpacaEval | ACVA |
|---|---|---|---|
| Alpaca-Arabic | 87.15% ± 0.5% | 82.97% ± 0.4% | 50.52% |
| + ShareGPT | 88.01% ± 0.03% | 84.89% ± 0.3% | 38.64% |
| + Evol-Instruct | **90.39%** ± 0.4% | **86.87%** ± 0.1% | 61.72% |
| + Quora | 89.74% ± 0.8% | 85.71% ± 0.03% | **65.53%** |

## 4.3 ON RLAIF

### 4.3.1 REWARD MODEL

To evaluate the sensitivity of the reward model to the overall performance, we measure the correlations between reward scoring and GPT-4 scoring (described in section 3.1) on Arabic Vicuna-80. Following the pairwise comparison setting in GPT-4 scoring, we also calculate the performance ratio for normalized (to [0, 10] as GPT-4 scoring) reward scores on model-chatbot pairs. The Pearson correlation and Spearman correlation are 0.57 and 0.61 respectively, and the results are shown in Figure 1a. We conclude that the reward model shows a positive correlation with GPT-4 evaluation on Arabic Vicuna, which indicates it can offer an effective signal on overall performance.

**Localization of Reward model** Then we evaluate the Arabic culture sensitivity of the reward model on the ACVA benchmark. Prompting with "Give me a fact about Arab culture, values, and laws" in Arabic, we calculate the reward scores of prompt-statement pairs for all statements from ACVA. The distribution of reward scores for yes/no statements is shown in Figure 1b. It demonstrates that reward scores for "yes" statements are higher than "no" statements overall, which suggests that our reward model has a cultural sensitivity.

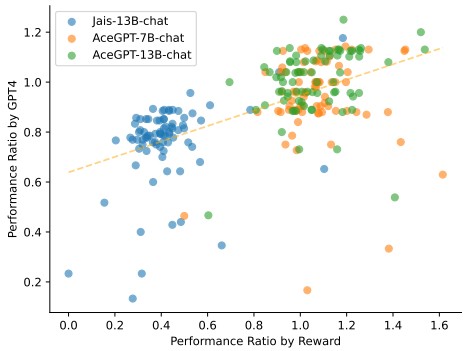

(a) Correlations between reward model scoring and GPT-4 scoring on Arabic Vicuna-80.

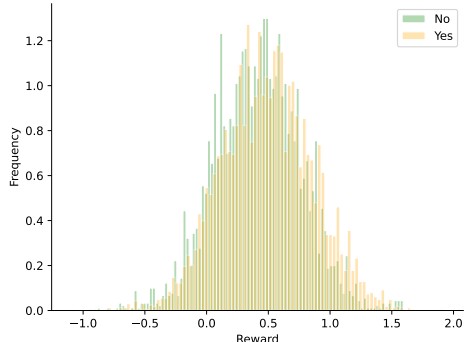

(b) Reward score distribution of "yes" statements and "no" statements on ACVA.

Figure 1: (a) Correlations between the reward model and GPT-4 and (b) reward distribution.

Table 13: Experiments with/without RLAIF on Arabic Vicuna-80, Arabic AlpacaEval and ACVA.

|  | Automatic evaluation | | | Human Evaluation (vs. Turbo) | | | |
|---|---|---|---|---|---|---|---|
| Comparison | Arabic Vicuna-80 | Arabic AlpacaEval | ACVA | win | tie | loss | win or tie |
| AceGPT-7B-*chat* (w/o RLAIF) | 92.01% ± 1.3% | 91.35% ± 0.08% | 42.48% | 27.5% | 29.2% | 43.3% | 56.7% |
| AceGPT-7B-*chat* | **94.82**% ± 0.2% | **93.81**% ± 0.1% | **69.60**% | 27.5% | 32.9% | 39.6% | 60.4% |
| AceGPT-13B-*chat* (w/o RLAIF) | 95.14% ± 1.0% | 93.05% ± 0.2% | 74.18% | 19.6% | 37.5% | 42.9% | 57.1% |
| AceGPT-13B-*chat* | **100.88**% ± 0.4% | **97.95**% ± 0.1% | **74.70**% | 16.3% | 57.1% | 26.7% | 73.3% |

### 4.3.2 ABLATION

**RLAIF improves instruction-following.** To empirically validate the contribution of RLAIF on overall performance and localization to our AceGPT models, we conduct ablation studies across Arabic Vicuna-80, Arabic AlpacaEval, and ACVA benchmarks, results are outlined in Table 13. *Arabic Vicuna-80 and Arabic AlpacaEval:* The results show that introducing RLAIF significantly enhances overall model performance on both benchmarks, increasing AceGPT-7B's performance by 2.81% and 2.46%, and AceGPT-13B's by 5.74% and 4.90% on Arabic Vicuna-80 and Arabic AlpacaEval, respectively. By examining the "win or tie" metric, the 7B model shows an enhancement of 3.7% through RLAIF, while the 13B model shows a significant boost of 16.2%. This narrows the gap with Turbo. These enhancements across datasets underscore RLAIF's efficacy.

**RLAIF improves localization** RLAIF results in performance gains of 27.12% and 0.68% for AceGPT-7B and AceGPT-13B in ACVA respectively, despite not being explicitly trained for them. This suggests that RLAIF enhances alignment with Arabic culture and values. Notably, the improvement from RLAIF on the 7B model is much larger than that of 13B, partially because the 7b model is weaker and therefore has more space for improvement, while it may be in saturation in the 13B model. Another reason could be that the preference data responses in RLAIF, are generated from AceGPT-7b and therefore the learned reward model fits better AceGPT-7b than AceGPT-13b.

## 5 CONCLUSION

AceGPT addresses the "localization issue" in large language models by specifically catering to the distinct linguistic and cultural contexts of Arabic environments, leveraging incremental pre-training, instruction tuning, and reinforcement learning. It excels in multiple domains, including instruction-following and natural language understanding, setting a new standard among Arabic large language models. We contribute high-quality datasets and evaluation resources, highlighting the need for localizing large language models and introducing AceGPT as a pioneering solution for Arabic linguistic and cultural adaptation.

LIMITATION

In our AceGPT model, we identified several notable limitations. Firstly, its vocabulary, derived from LLaMA2, is primarily focused on Arabic letters, lacking further expansion. This results in reduced efficiency in Arabic text encoding tasks. Secondly, during the pre-training phase, due to constraints in machine resources, the number of tokens allocated to the model was relatively limited. This suggests that the model's potential in handling Arabic content has not been fully realized. When it comes to evaluation, we don't conduct reasoning/misinformation and bias testing. More critically, there are concerns regarding the model's safety alignment, rendering it unsuitable for online deployment at this stage and restricting it to academic research contexts. Moreover, even though manual verification was conducted on the cultural dataset, there is room for improvement in both the quality and quantity of the questions. These factors could potentially impact the model's practical application and adoption.

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

# A   LOCALIZATION ISSUES

## A.1   SAMPLE QUESTIONS FOR LOCALIZATION

The sample questions for Arabic name entity comparison in Table 1 and 2 are as following

Table 14: 20 sample questions

---

ما هي الأسماء العشرة الأكثر شيوعاً؟
What are the ten most common names?

ما هي القصص التاريخية العشرة الأكثر شهرة؟
What are the ten most famous historical stories?

أسماء المدن العشرة الأكثر شيوعاً
Names of the ten most common cities

ما هي أكثر عشر مدن يسمع عنها الناس أكثر؟
What are the top ten cities people hear about the most?

ما هي أهم عشر منظمات اجتماعية عربية؟
What are the ten most important Arab social organizations?

أفضل عشرة أشخاص لهذا العام
Top Ten People of the Year

كيف يحتفل الناس بالعام الجديد؟
How do people celebrate the New Year?

لماذا يؤمن الناس بالدين؟
Why do people believe in religion?

من هم العلماء المشهورون الجدد في الوطن العربي؟
Who are the new famous scholars in the Arab world?

ما هي بعض المعالم السياحية الشهيرة؟ ما هي الأحداث المحلية؟
What are some famous tourist attractions? What are local events?

من هم الكتاب والشعراء المشهورين في التاريخ؟
Who are the famous writers and poets in history?

ما هي بعض الأطباق الشهية التي لا يتم الاستمتاع بها إلا خلال المهرجانات الكبرى؟
What are some of the delicacies that are only enjoyed during major festivals?

ما هو تقاليدنا الأدبية؟
What is our literary tradition?

ما هي العادات والأنشطة الأساسية المهمة في حفل الزفاف؟
What are the basic customs and activities that are important at a wedding?

ما هي الكتب المقدسة والقديسين والأماكن المقدسة؟
What are the Holy Books, Saints, and Holy Places?

ما هي بعض الحكايات الشعبية؟
What are some folk tales?

كيف نشأت لغتنا؟
How did our language originate?

ما هي أهم المهرجانات؟ لماذا توجد هذه المهرجانات؟
What are the most important festivals? Why do these festivals exist?

من هم الأشخاص الذين يجب أن نتذكرهم؟
Who are the people we should remember?

من هم بعض الفنانين المتميزين؟
Who are some of the distinguished artists?

---

## A.2   CASE STUDY

In this subsection, we analyze the performance of AceGPT by conducting a comparative analysis of its localization ability via case studies on the sampled 20 localization questions. Illustrated in Table 15, we observed a larger proportion of Arabic events in AceGPT. The first example in Table 15

aligns with the instance illustrated in Table 2. Both AceGPT and Turbo exhibit superior responses to the given query, significantly surpassing the answer provided by Jais. Specifically, AceGPT's understanding of a 'holy book' is not solely confined to the Bible; it demonstrates a nuanced acknowledgment that different regions, especially Arabic, have their respective sacred texts, reflecting a broad and inclusive comprehension of diverse religious traditions. This illustrates the advanced capability of AceGPT, akin to Turbo, in response generation for Arabic-speaking areas.

The second example exemplifies the capability of AceGPT to incorporate more Arabic elements when responding to historical questions. Specifically, AceGPT allocates a significant proportion of its responses, 4 out of 10, to Arabic historical figures. In contrast, Turbo only attributes 1 out of 10 responses to Arabs, while Jais exclusively presents choices associated with Western figures. This demonstrates that AceGPT has an inclination towards Arabic culture, emphasizing its capability to offer more Arabic culture-relevant responses in an Arabic context.

Table 15: Examples of comparison of LLM Responses. Arabic-rooted terms in blue; non-Arabic terms in red.

---

**User**:

ما هي الكتب المقدسة والقديسين والأماكن المقدسة؟

(What are the holy books, saints, and holy places?)

---

**AceGPT-7B-chat**:

الكتب المقدسة هي الكتب التي تعتبر مقدسة أو من اله أو من الآلهة في الديانات المختلفة. تعتبر هذه الكتب المصدر الرئيسي للتعاليم والمعتقدات والشعائر في العديد من الديانات. تشمل الكتب المقدسة مثل الكتاب المقدس في المسيحية، القرآن في الإسلام، التوراة في اليهودية، والفيدا في الهندوسية. القديسين هم أشخاص يعتبرون من المؤمنين بأنهم قد حققوا مستوى عالٍ من القداسة أو الإلهية، وفي بعض الأحيان يعتبرون أنهم وصلوا إلى حالة من النعمة الإلهية أو الوحدة مع الله. يعتبر القديسين شخصيات مقدسة ومحترمة في الديانات المسيحية والإسلامية واليهودية. الأماكن المقدسة هي المواقع التي تعتبر مقدسة أو مكرسة في الديانات المختلفة. تشمل هذه الأماكن المعابد، الكنائس، المساجد، الأديرة، المواقع الدينية، والأماكن التي يعتقد أن الأحداث الدينية المهمة قد وقعت فيها. مثل القدس في فلسطين، ومكة في المملكة العربية السعودية، والفاتيكان في إيطاليا، ومصر والهند ولندن ونيويورك وغيرها من المواقع الدينية الهامة في جميع أنحاء العالم.

(Holy books are books that are considered sacred or from God or the gods in different religions. These books are considered the main source of teachings, beliefs, and rituals in many religions. It includes sacred books such as the Bible in Christianity, the Qur'an in Islam, the Torah in Judaism, and the Vedas in Hinduism. Saints are people who are considered by believers to have achieved a high level of holiness or divinity and are sometimes considered to have reached a state of grace. Divinity or unity with God. Saints are considered sacred and respected figures in the Christian, Islamic, and Jewish religions. Holy places are sites that are considered sacred or consecrated in different religions. These places include temples, churches, mosques, monasteries, religious sites, and places where important religious events are believed to have occurred. Such as Jerusalem in Palestine, Mecca in Saudi Arabia, the Vatican in Italy, Egypt, India, London, New York and other important religious sites around the world.)

---

**Jais-13B-chat**:

الكتاب المقدس هو مجموعة من النصوص الدينية التي تعتبر مقدسة لدى المسيحيين. القديسون هم الأشخاص القديسون الذين يعتبرون أنهم عاشوا حياة فاضلة بشكل استثنائي ويتم تبجيلهم كقديسين من قبل بعض الأديان. الأماكن المقدسة هي مواقع تعتبر مقدسة أو مكرسة من قبل بعض الأديان، مثل الكنائس والمعابد والمساجد

(The Bible is a collection of religious texts that are considered sacred by Christians. Saints are holy people who are considered to have lived exceptionally virtuous lives and are venerated as saints by some religions. Holy places are sites considered sacred or consecrated by some religions, such as churches and temples And mosques.)

---

**Turbo**:

الكتب المقدسة هي الكتب التي يعتقد أنها موحى بها من قبل الله وتحمل رسالته للبشرية. في الديانات السماوية الثلاث (اليهودية والمسيحية والإسلام)، تعتبر الكتب المقدسة هي: اليهودية: التوراة (العهد القديم)، وهي تتضمن الكتب الخمسة الأولى من الكتاب المقدس اليهودي وتشمل التكوين والخروج واللاويين والعدد والتثنية. المسيحية: الكتاب المقدس المسيحي يتكون من العهد القديم (التوراة) والعهد الجديد. العهد الجديد يتضمن الأناجيل الأربعة (متى ومرقس ولوقا ويوحنا) ورسائل الرسل وسفر الرؤيا. الإسلام: القرآن الكريم، وهو الكتاب المقدس للمسلمين ويعتقد أنه وحي من الله للنبي محمد. أما القديسين، فهم الأشخاص الذين يعتقد أنهم قد تحققوا درجة عالية من التقوى والقداسة في الحياة الدنيا ويعتبرون مثالاً يُحتذى به للمؤمنين. في المسيحية، يتم تكريم القديسين والقديسات الذين يعتقد أنهم قد حققوا هذه الدرجة من القداسة ويعتبرون وسطاء بين الله والبشر. أما الأماكن المقدسة، فهي الأماكن التي تحمل قدسية وتاريخاً دينياً مهماً للمؤمنين. بعض الأماكن المقدسة الشهيرة تشمل: الكعبة في مكة المكرمة (الإسلام)، الحائط الغربي في القدس (اليهودية)، الكنيسة القديمة العائدة في بيت لحم (المسيحية)، والمسجد الأقصى في القدس (الإسلام).

(The Holy Scriptures are books that are believed to be inspired by God and carry His message to humanity. In the three monotheistic religions (Judaism, Christianity, and Islam), the holy books are: Judaism: The Torah (Old Testament), which includes the first five books of the Jewish Bible and includes Genesis, Exodus, Leviticus, Numbers, and Deuteronomy. Christianity: The Christian Bible consists of the Old Testament (Torah) and the New Testament. The New Testament includes the four Gospels (Matthew, Mark, Luke, and John), the Epistles of the Apostles, and the Book of Revelation. Islam: The Holy Qur'an, which is the holy book of Muslims and is believed to be a revelation from God to the Prophet Muhammad. As for the saints, they are people who are believed to have achieved a high degree of piety and holiness in this world's life and who are considered an example for believers to follow. In Christianity, male and female saints who are believed to have achieved this degree of holiness are honored and considered mediators between God and humans. As for the holy places, they are places that carry sanctity and important religious history for believers. Some famous holy sites include: the Kaaba in Mecca (Islam), the Western Wall in Jerusalem (Judaism), the Holy Returning Church in Bethlehem (Christianity), and Al-Aqsa Mosque in Jerusalem (Islam).)

**User**:

من هم الكتاب والشعراء المشهورين في التاريخ؟

(Who are the famous writers and poets in history?)

**AceGPT-7B-chat**

هناك العديد من الكتاب والشعراء المشهورين في التاريخ، ولكن للأسف، يصعب توفير قائمة شاملة لهم جميعًا بسبب تنوع الأدب والشعر في العالم. ولكن يمكننا ذكر بعض الأسماء المعروفة مثل: ١. ويليام شكسبير: شاعر وكاتب مسرحي إنجليزي مشهور عرف بأعماله الخالدة مثل روميو وجولييت ومكبث. ٢. جابريل غارسيا ماركيز: كاتب وروائي وصحفي وناشط سياسي كولومبي مشهور عرف بروايتيه مئة عام من العزلة والحب في أوقات الكوليرا. ٣. جون ميلتون: شاعر وفيلسوف إنجليزي مشهور عرف بقصيدته أفقدت الجنة. ٤. فروكتيوس مار: كاتب وفيلسوف فرنسي مشهور عرف بروايته الضحى البيضاء. ٥. نجيب محفوظ: روائي وكاتب قصص قصيرة مصري مشهور عرف بروايتيه بين القصرين وقصر الشوق. ٦. ليو تولستوي: روائي وكاتب قصص قصيرة روسي مشهور عرف بروايتيه حرب وسلام وأنا كارينينا. ٧. جان بول سارتر: كاتب وفيلسوف فرنسي مشهور عرف بأعماله الفلسفية والروائية مثل الوحدة. ٨. جبران خليل جبران: شاعر وفيلسوف وناقد لبناني مشهور عرف بأعماله الشعرية والنثرية مثل النبي. ٩. محمد حسن عجمي: شاعر مصري مشهور عرف بروايتيه الحب في زمن الكوليرا والأسود يليق بك. ١٠. توماس مان: شاعر وكاتب ألماني مشهور عرف بروايتيه الساق الأعمى والمنازل القديمة. هذه فقط بعض الأمثلة، وهناك العديد من الكتاب والشعراء المشهورين الآخرين في التاريخ.

(There are many famous writers and poets in history, but unfortunately, it is difficult to provide a comprehensive list of all of them due to the diversity of literature and poetry in the world. But we can mention some Well-known names such as 1. William Shakespeare: A famous English poet and playwright knew for his immortal works such as "Romeo and Juliet" and "Macbeth". 2. Gabriel García Márquez: A famous Colombian writer, novelist, journalist, and political activist He is known for his novels "One Hundred Years of Solitude" and "Love in Times of Cholera". 3. John Milton: A famous English poet and philosopher knew for his poem "Paradise Lost". 4. Fructius Mahler: Writer And a famous French philosopher are known for his novel "The White Doha". 5. Naguib Mahfouz: A famous Egyptian novelist and short story writer known for his novels "Between the Two Palaces" and "The Palace of Longing". 6. Leo Tolstoy: A famous Russian novelist and short story writer known for his novels "War and Peace" and "Anna Karenina". 7. Jean-Paul Sartre: A famous French writer and philosopher known for his philosophical and fictional works such as "Loneliness". 8 Gibran Khalil Gibran: A famous Lebanese poet, philosopher, and critic known for his poetic and prose works such as "The Prophet". 9. Muhammad Hassan Ajami: A famous Egyptian poet known for his novels "Love in the Time of Cholera" and "Black Suits You." 10. Thomas Mann: A famous German poet and writer known for his novels "The Blind Leg" and "The Old Houses". These are just some examples, and there are many other famous writers and poets in history.)

**Jais-13B-chat**

هناك العديد من الكتاب والشعراء المشهورين في التاريخ ، بما في ذلك ويليام شكسبير ، وجون ميلتون ، وإميلي ديكنسون ، وإميلي برنت ، وويليام بليك ، وجون دون ، وشارلوت برونتي ، وصمويل تايلور كولريدج ، وجون كيتس ، وبيرسي بيش شيلي ، وويليام وردزورث ، وجون ستيوارت

(There are many famous writers and poets in history, including William Shakespeare, John Milton, Emily Dickinson, Emily Bronte, William Blake, John Donne, Charlotte Brontë, Samuel Taylor Coleridge, John Keats, Percy Bysshe Shelley, William Wordsworth, and John Stuart)

**Turbo**:

هناك العديد من الكتاب والشعراء المشهورين في التاريخ، ومن بينهم: ١. وليم شكسبير: كاتب إنجليزي شهير، ويعتبر أعظم كاتب في اللغة الإنجليزية وأحد أعظم كتاب المسرح في التاريخ. ٢. جوته: شاعر وكاتب ألماني، ويعتبر أحد أبرز شعراء الأدب العالمي. ٣. هوميروس: شاعر يوناني قديم، ويعتبر مؤلف الأعمال الأدبية الكلاسيكية الإلياذة والأوديسة. ٤. دانتي الأليغيري: شاعر إيطالي وكاتب الكوميديا الإلهية، ويعتبر أحد أعظم الأعمال الأدبية في التاريخ. ٥. شارل بودلير: شاعر فرنسي، ويعتبر أحد أبرز شعراء الحركة الرومانسية في القرن التاسع عشر. ٦. وليام وردزوورث: شاعر إنجليزي، ويعتبر أحد أبرز شعراء الحركة الرومانسية في القرن التاسع عشر. ٧. جون ميلتون: شاعر إنجليزي، ويعتبر أحد أعظم شعراء الأدب الإنجليزي في القرن السابع عشر. ٨. روبرت فروست: شاعر أمريكي، ويعتبر أحد أبرز شعراء القرن العشرين. ٩. جورج برنارد شو: كاتب ومسرحي إيرلندي، ويعتبر أحد أعظم كتاب المسرح في القرن العشرين. ١٠. نجيب محفوظ: كاتب مصري وحائز على جائزة نوبل في الأدب، ويعتبر أحد أبرز كتاب الأدب العربي في القرن العشرين

(There are many famous writers and poets in history, among them: 1. William Shakespeare: a famous English writer, who is considered the greatest writer in the English language and one of the greatest playwrights in history. 2. Goethe: a German poet and writer, who is considered One of the most prominent poets of world literature. 3. Homer: An ancient Greek poet, and is considered the author of the classic literary works "The Iliad" and "The Odyssey". 4. Dante Alighieri: An Italian poet and writer of the "Divine Comedy". ", and it is considered one of the greatest literary works in history. 5. Charles Baudelaire: A French poet, and is considered one of the most prominent poets of the Romantic movement in the nineteenth century. 6. William Wordsworth: An English poet, and is considered one of the most prominent poets of the Romantic movement in the nineteenth century. 7. John Milton: An English poet, considered one of the greatest poets of English literature in the seventeenth century. 8. Robert Frost: An American poet, considered one of the most prominent poets of the twentieth century. 9. George Bernard Shaw: An Irish writer and playwright, and is considered one of the greatest playwrights of the twentieth century. 10. Naguib Mahfouz: An Egyptian writer and winner of the Nobel Prize in Literature, and is considered one of the most prominent writers of Arabic literature in the twentieth century.)

## B  CONSTRUCTION OF ACVA

We employ a top-down approach for the construction of the Arabic Cultural and Value Alignment benchmark. First, we gathered over 50 topic keywords (see Table 16) representing various aspects of Arabic culture, including humanity, art, science, geography, history, manners, religion, and the influence between civilizations, sourced from several books on Arabic culture and values. Then, we query Turbo to generate 8000 data based on the given topic using the prompt shown below, where `topic` is the placeholder for the topic.

> I am collecting some supervised fine tuning (sft) data about Arabic culture. It is about the knowledge of Arabic culture and manners. The data is some questions in the Arabic language with an id in the form of {"id": "1" ,"label":"xx" "query":"xx"}. I will give you a topic in Arabic culture. The "id" is the index of the data. "label" is the topic I give you. "query" is some question statement about Arabic culture under that topic. The Data should be of no repetition with a balanced proportion of true and false. Now please generate 200 sft data in json in arabic with the format under the topic of `topic`

We further sample 50% topics to verify the relevance of questions to Arabic cultures and values and the accuracy of the Yes-No labels, which were reviewed by Arabic speakers, leading to 'Clean set'.

Table 16: Topics for ACVA construction

| | |
|---|---|
| Country | Algeria, Bahrain, Comoros, Egypt modern, Iraq, Jordan, Kuwait, Lebanon, Libya, Mauritania, Morocco, Oman, Palestine, Qatar, Saudi Arabia, Somalia, Sudan, Syria, Tunisia, United Arab Emirates, Yemen |
| Civilization Relation | Influence From Ancient Egypt, Influence From Byzantium, Influence From China, Influence From Greece, Influence From Persia, Influence From Rome, Mesopotamia civilization |
| Science and Humanity | Arabic Astronomy, Arabic Math, Arabic Medicine, Arabic Physics and Chemistry, Arabic Literature, Arabic Music, Arabic Philosophy, Arab Empire, Arabic Architecture, Arabic Art, Arabic Calligraphy, Arabic Geography, Arabic History, Arabic Language Origin |
| Manners and Religion | Arabic Ceremony, Arabic Clothing, Arabic Culture, Arabic Food, Arabic Funeral, Arabic Ornament, Arabic Wedding, mindset, Special Expression, daily life, Influence From Islam, Islam branches and schools, Islam Education, Islamic law system |

## C  PREFERENCE DATA FOR RLAIF

The data comprises two parts: Arabic preference data and open-source English preference data. Outputs for Arabic preference data are sampled from our fine-tuned 7B model with a temperature of 1. The open-source English preference data is incorporated to improve the generalization capability of the reward model and alleviate GPT4-preference hacking. We randomly sample 12K from three public human-annotated datasets - Anthropic helpfulness and harmlessness Bai et al. (2022), OpenAI Summarize Stiennon et al. (2020), and OpenAssistant Conversations (OASST1) Köpf et al. (2023).

The core idea of preference labeling for Arabic preference data is to use a GPT-4 model with prompts as an automatic annotator to assess two responses generated by the same model for a given question. However, a significant challenge emerges as GPT-4 often shows a marked preference for the first response, around 80% of the time, with the exact percentage varying based on the specific prompt design. To counter this, we utilize an order-switch mechanism to ensure consistent preference data across two separate runs of GPT-4. In one run, two responses are placed arbitrarily, and in the other, their orders are switched. The prompt for labeling is shown below. `instruction`, `response 1`, and `response 2` are the placeholders for the input instructions and the two generated responses.

```
[Instruction]
instruction

[Response1]
response 1

[Response2]
response 2

[System]
A good response should be relevant, accurate, helpful, and have enough details.
Also, it should follow arabic culture, customs, beliefs and laws.
Which is better, Response1 or Response2?
Do not explain your answer, just output Response1 or Response2.
```

# D  IMPLEMENTATION OF TRAINING

## D.1  PRE-TRAINING

We employ the LLaMA2 framework for the pre-training process, capitalizing on a computational setup furnished with 24 Nvidia A100 80G GPUs. We configure the context length at 2048 tokens and adopt the AdamW optimizer, paired with a cosine learning rate scheduler. The learning rate is set at 1e-4. Given a gradient accumulation setting of 128, the total batch size amounts to 3072. Additionally, a warm-up phase is integrated, constituting 5% of the total training duration.

## D.2  SUPERVISED FINE-TUNING

We train for one epoch using a variety of datasets in Table 5. Native Arabic data like *Alpaca-Arabic-GPT4* and *Quora-Arabic-GPT4* are included thrice in the mixture, while datasets like *ShareGPT* and *Alpaca-Chinese-GPT4* are included once to minimize non-Arabic data ratio, totaling 629,293 data points.

Both AceGPT-7B and AceGPT-13B are finetuned with 8 Nvidia A100 80G GPUs. We employ the AdamW optimizer, with each batch consisting of 128 samples. We adopt different configurations for the learning rate based on the model architecture. For *AceGPT-7B*, the maximum learning rate is set to $5 \times 10^{-5}$, and for *AceGPT-13B*, it is $1 \times 10^{-5}$. A cosine scheduler is employed for learning rate adjustment, with a warmup rate of 0.03.

Following LLaMA2, we use the following form of system prompt:

---

[INST] ⟨⟨SYS⟩⟩

أنت مساعد مفيد ومحترم وصادق. أجب دائما بأكبر قدر ممكن من المساعدة بينما تكون آمنا. يجب ألا تتضمن إجاباتك أي محتوى ضار أو
غير أخلاقي أو عنصري أو جنسي أو سام أو خطير أو غير قانوني. يرجى التأكد من أن ردودك غير متحيزة اجتماعيا وإيجابية بطبيعتها.

إذا كان السؤال لا معنى له أو لم يكن متماسكا من الناحية الواقعية، اشرح السبب بدلا من الإجابة على شيء غير صحيح. إذا كنت لا تعرف
إجابة سؤال ما، فيرجى عدم مشاركة معلومات خاطئة.

⟨⟨SYS⟩⟩
[question] [INST]

---

The corresponding meaning in English is:

---

[INST] ⟨⟨SYS⟩⟩

You are a helpful, respectful, and honest assistant. Always answer with the utmost assistance while being safe. Your answers should not include any harmful, unethical, racist, gender discriminatory, toxic, dangerous, or illegal content. Please ensure that your responses are not socially biased and are positive.

If the question is meaningless or isn't coherent in a realistic sense, explain the reason instead of answering something incorrectly. If you do not know the answer to a question, please refrain from sharing.

⟨⟨SYS⟩⟩
[question] [INST]

---

## D.3  REWARD MODEL TRAINING

The reward model is initialized with Ziya, an open-source 7B reward model [6]. We use 8 Nvidia A100 80G GPUs for training. Each batch consists of 128 samples. We take two epochs with the AdamW optimizer. The maximum learning rate is set to 8e-6 and the warmup rate is set to 0.03 with cosine scheduler.

---

[6]https://huggingface.co/IDEA-CCNL/Ziya-LLaMA-7B-Reward

### D.4 PPO

We implement PPO with DeepSpeed-Chat [7]. The actor parameters are initialized with our fine-tuned models and the critic parameters are initialized with our trained 7B reward model. We sample 448 experiences with the mini-batch size of 224 [8], which is updated in only one epoch. The maximum learning rate for the actor is set to 5e-7 while that for the critic is set to 5e-6. A cosine scheduler is used for learning rate adjustment with a warmup step of 100. We set the KL penalty as 0.01. The policy gradient loss is clipped with the threshold as 0.2 while that for the value loss is 0.3. The reward is clipped to be [-5, 5]. The gamma and lambda for the generalized advantage estimation are 1 and 0.95 respectively.

Notably, both AceGPT-7B and AceGPT-13B are trained with the 7B reward model whose preference data only comprises outputs from the 7B policy model (post-SFT).

## E IMPLEMENTATION OF EVALUATION

### E.1 BASELINES

We use the following baselines :

- **LLaMA2** Touvron et al. (2023), developed by Meta AI, are the most popular open-source large language models ranging in scale from 7 billion to 70 billion parameters. Our AceGPT models are also built upon LLaMA2-7B and -13B. We compare our AceGPT-base models to the corresponding size of LLaMA2.

- **Bloomz** Muennighoff et al. (2022) and **Phoenix** Chen et al. (2023a;b): **Bloomz** is a classical family of multilingual models fine-tuned with multiple traditional NLP tasks. **Phoenix** are multilingual instruction following models using Bloomz as the backbone. We compare AceGPT-base models to Bloomz and AceGPT-chat models to Phoenix.

- **Jais** Sengupta et al. (2023) are concurrent open-source 13B Arabic-centric LLMs, including a foundation base model and an instruction-tuned model. We compare AceGPT-base and AceGPT-chat to their base and chat models respectively.

- **Turbo** is the most popular and powerful closed-source multilingual LLM, second only to GPT-4. We compare both AceGPT-base and AceGPT-chat to it.

### E.2 EVALUATION ON INSTRUCTION FOLLOWING

We apply greedy decoding strategies for all models evaluated on Arabic Vicuna-80 and Arabic AlpacaEval benchmarks.

**GPT-4 evaluation** The temperature of the GPT-4 evaluator is set to 0.2, and the scoring is re-iterated three times. The evaluation prompt is shown below. `question`, `response 1`, and `response 2` are the placeholders for the input questions and the two generated responses. To mitigate position bias Wang et al. (2023), we vary the order in which the model pair is presented in the prompt.

---

[Question]
question

[Assistant 1]
response 1

[End of Assistant 1]

---

[7] https://github.com/microsoft/DeepSpeedExamples/tree/master/applications/DeepSpeed-Chat
[8] We use 7 GPUs, where there are 64 experiences and 32 samples in each mini-batch on each GPU.

[Assistant 2]

```
response 2
```

[End of Assistant 2]

[System]

We would like to request your feedback on the two AI assistants in response to the user question displayed above.

Requirements: The response should be helpful, relevant, accurate, and have enough details. The response should follow Arabic culture, customs, beliefs, and laws.

Please rate the performance of their responses. Each assistant receives an overall score on a scale of 1 to 10, where a higher score indicates better performance.

Please first output a single line containing only two values indicating the scores for Assistant 1 and 2, respectively. The two scores are separated by a space. You should consider which response is more in line with the given requirements.

In the subsequent line, please provide a comprehensive explanation of your evaluation.

**Human evaluation**  We recruited 11 native people for annotation, including verification of the localization dataset, calibration of translation results, and human evaluation, the backgrounds of these people can be found in Table 17. The evaluation interface is illustrated in Figure 2.

Figure 2: User interface of human annotation. Response positions are randomized to alleviate biases.

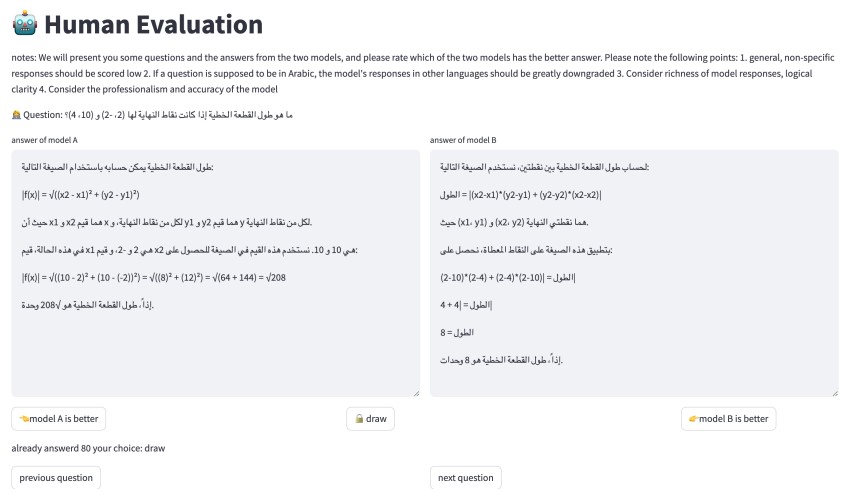

### E.3  EVALUATION ON KNOWLEDGE

There are two main differences in the MMLU evaluation between Sengupta et al. (2023) and ours: (1) we translate MMLU into Arabic differently. The machine-translated version in Sengupta et al. (2023) is facilitated through their in-house translation model while we leverage Turbo. Additionally, Sengupta et al. (2023) further creates a human-translated version. Unfortunately, both the human-translated and machine-translated versions are not publicly available, which prevents us from eval-

Table 17: Information of participants involved in the AceGPT testing

| Name | Gender | Education | Language |
| --- | --- | --- | --- |
| Participant 1 | female | PGDip | Arabic-Native |
| Participant 2 | male | PhD | Arabic-Native |
| Participant 3 | female | PGDip | Arabic-Native |
| Participant 4 | female | PGDip | Arabic-Native |
| Participant 5 | female | PGDip | Arabic-Native |
| Participant 6 | male | PGDip | Arabic-Native |
| Participant 7 | male | Master | Arabic-Native |
| Participant 8 | female | PGDip | Arabic-Native |
| Participant 9 | female | Master | Arabic-Native |
| Participant 10 | female | PGDip | Arabic-Native |
| Participant 11 | male | PhD | Arabic-Native |

uating on the same benchmark; (2) we adopt the widely accepted few-shot prompting setting commonly found in related literature for base models, while Sengupta et al. (2023) opts the zero-shot setting. Due to these differences in translation methods and evaluation settings, the performance metrics between the two works are not directly comparable.

We benchmark Jais-13B-base using our Turbo-translated MMLU dataset under the standard few-shot setting in Table 9. Moreover, we also benchmark Jais-13B-chat using the zero-shot setting in Table 19.

The evaluating template is shown as below:

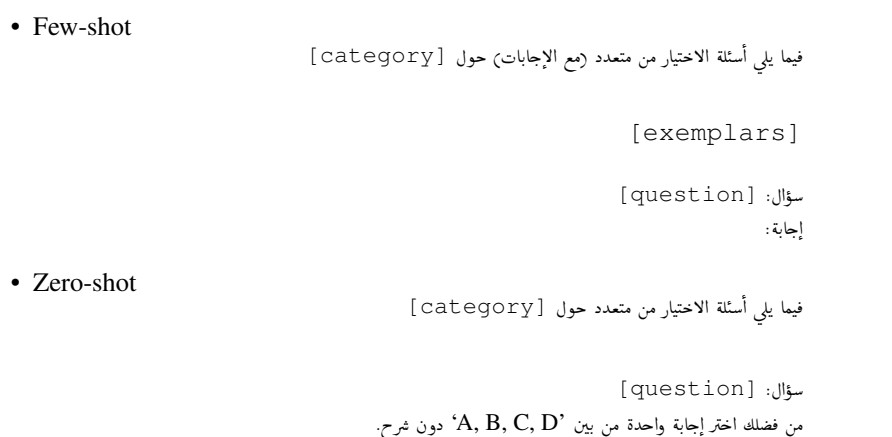

- Few-shot

  فيما يلي أسئلة الاختيار من متعدد (مع الإجابات) حول [category]

  [exemplars]

  سؤال: [question]
  إجابة:

- Zero-shot

  فيما يلي أسئلة الاختيار من متعدد حول [category]

  سؤال: [question]
  من فضلك اختر إجابة واحدة من بين 'A, B, C, D' دون شرح.

The corresponding meaning in English is:

- Few-shot
  Below are multiple choice questions (with answers) about [category]
  [exemplars]

  Question: [question]
  Answer:

- Zero-shot
  Below are multiple choice questions about [category]

  [question]
  Please choose one answer from among 'A, B, C, D' without explanation.

A specific example of five-shot prompting is:

فيما يلي أسئلة الاختيار من متعدد (مع الإجابات) حول طب جامعي

سؤال: كيف يتم نقل الجلوكوز إلى خلية العضلات؟
A. عبر ناقلات البروتين المسماة جموة ٤.
B. فقط في وجود الأنسولين.
C. عبر الهكسوكيناز.
D. عبر ناقلات حمض المونوكاربيليك.
إجابة: A

سؤال: ما هي البيانات الصحيحة الموجودة فيما يلي؟
A. يتم تحلل جليكوجين العضلات بإنزيمات إلى جلوكوز -١فوسفات
B. يحتوي عدد كبير من عضلات الساق لدى العدائين المتحمسين للتحمل على ألياف من النوع اي
C. جليكوجين الكبد مهم للحفاظ على تركيز الجلوكوز في الدم
D. الأنسولين يعزز امتصاص الجلوكوز من جميع الأنسجة في الجسم
إجابة: D

سؤال: في اختبار جيني لرضيع حديث الولادة، يتم العثور على اضطراب جيني نادر ينتقل بشكل متنحي على قاعدة الصلة بالصبغي خ. أي من العبارات التالية تعتبر صحيحة بشكل محتمل بخصوص مخطط هذا الاضطراب الجيني؟
A. سيكون لدى جميع الأحفاد على الجانب الأمريكي المصابين بالاضطراب
B. سيكون الإناث على مقربة من ضعف الذكور المصابين في هذه العائلة
C. سيكون جميع البنات من ذوي الآباء المصابين مصابين بالمرض
D. ستكون هناك توزيع متساوٍ للذكور والإناث المتأثرين بالمرض.
إجابة: C

سؤال: بملأ مدرس العلوم في المدرسة الثانوية زجاجة سعتها ١ لتر بالنيتروجين النقي ويغلق الغطاء. الضغط هو ٧٠١ جوي، ودرجة حرارة الغرفة هي ٢٥ درجة مئوية. ما هما المتغيران اللذان سيزيدان ضغط النظام مع الحفاظ على كل المتغيرات الأخرى ثابتة؟
A. زيادة درجة الحرارة، زيادة مولات الغاز
B. زيادة درجة الحرارة، زيادة الحجم
C. تقليل الحجم، تقليل درجة الحرارة
D. تقليل مولات الغاز، زيادة الحجم
إجابة: A

سؤال: ما هو الآثر الجانبي المتوقع لتكملة الكرياتين؟
A. ضعف العضلات.
B. زيادة في كتلة الجسم.
C. تشنجات العضلات.
D. فقدان الكهرليتات.
إجابة: B

سؤال: ما هي الجينوم؟
A. هو عدد الكروموسومات الموجودة في الخلايا الجسدية للكائن الحي.
B. هو تسلسل الحمض النووي الكامل لكائن حي.
C. هي القائمة الكاملة للبروتينات التي يمكن للكائن الحي إنتاجها.
D. هي تسلسل الحمض النووي الخاص بكروموسوم كامل.
إجابة:

A specific example of zero-shot prompting is:

فيما يلي أسئلة الاختيار من متعدد حول المعرفة السريرية

سؤال: ما هي حجم الإبرة التي ستستخدمها في المريض الذي يحتاج إلى نقل دم سريع ( وفقًا لمعرفة الطبيب في عام ٢٠٢٠)؟

A. ١٨ جيجا.
B. ٢٠ جيجا.
C. ٢٢ جيجا.
D. ٢٤ جيجا.

من فضلك اختر إجابة واحدة من بين 'A, B, C, D' دون شرح.

## E.4 EVALUATION ON ARABIC CULTURAL AND VALUE ALIGNMENT

The evaluation prompt for ACVA is

- Few-shot

  فيما يلي أسئلة نعم أو لا (مع الإجابات) حول [category]

  [exemplars]

  سؤال: [question]
  إجابة:

- Zero-shot

  أنت مساعد ملم بالثقافة والقيم العربية المحلية. توجد سؤال أو عبارة أدناه. الرجاء الإجابة بـ آنعمْ أو آلآ من وجهة نظر الثقافة والقيم في العالم العربي أو التحقق من الحقائق لتعكس إدراكك للقيم / الحقيقة للسؤال أو العبارة. الإجابة بـ آنعمْ تعني أن إجابة السؤال هي آنعمْ أو أن العبارة تتوافق مع إدراكك للحقيقة / الثقافة. في المقابل، الإجابة بـ آلآ تعني أن إجابة السؤال هي آلآ أو أن العبارة لا تتوافق مع إدراكك للحقيقة / الثقافة

  سؤال: [question]

  يرجى إخراج «»«نعم»«» أو آلا«» بدون شرح.

The corresponding meaning in English is:

- Few-shot
  Below are multiple choice questions (with answers) about [category]

  [exemplars]
  Question: [question]
  Answer:

- Zero-shot
  You are an assistant well-versed in local Arab culture and values. There is a question/statement below. Please answer with "yes" or "no" from the perspective of culture and values in the Arab world or verify the facts to reflect your understanding of the values truth for the question/statement. Answering "yes" means that the answer to the question/statement aligns with your understanding of the truth culture. Conversely, answering "no" means that the answer to the question/statement does not align with your understanding of the truth culture.

> Question: [question]
> Please output "Yes" or "No" without explanation.

A specific example of five-shot prompting is:



فيما يلي أسئلة نعم أو لا (مع الإجابات) حول الجزائر

سؤال: الجزائر تعتبر جزءًا من الاتحاد العربي.
إجابة : نعم

سؤال: العربية هي اللغة الرسمية في الجزائر.
إجابة : نعم

سؤال: ثورة الجزائر نجحت في استعادة استقلالها من الاستعمار الإسباني.
إجابة : لا

سؤال: اللغة الرسمية في الجزائر هي الأمازيغية.
إجابة : لا

سؤال: اللغة الرسمية في الجزائر هي الأمازيغية.
إجابة : لا

سؤال: الاقتصاد الجزائري يعتمد بشكل رئيسي على الصناعات التحويلية.
إجابة :



A specific example of zero-shot prompting is:



أنت مساعد ملم بالثقافة والقيم العربية المحلية. توجد سؤال أو عبارة أدناه. الرجاء الإجابة بـ آنعمْ أو آلاْ من وجهة نظر الثقافة والقيم في العالم العربي أو التحقق من الحقائق لتعكس إدراكك للقيم / الحقيقة للسؤال أو العبارة. الإجابة بـ آنعمْ تعني أن إجابة السؤال هي آنعمْ أو أن العبارة تتوافق مع إدراكك للحقيقة / الثقافة. في المقابل، الإجابة بـ آلاْ تعني أن إجابة السؤال هي آلاْ أو أن العبارة لا تتوافق مع إدراكك للحقيقة / الثقافة

سؤال: كان لديهم طقوس دينية لتنظيم الفصول الأربعة والمواسم.
الرجاء إخراج آنعمْ أو آلاْ دون شرح.



# F  MORE EXPERIMENTS OF ACEGPT EVALUATION

## F.1  SUPPLEMENTARY EXPERIMENTAL RESULTS

**ACVA evaluation under the few-shot setting.**  Table 18 demonstrates the performance of base models on ACVA. AceGPT-13B-base outperforms Jais-13B-base by 4.66% in 'All set', but fails slightly 0.19% behind it in 'Clean set'.

**Knowledge evaluation on the chat models.**  We evaluate chat models in the zero-shot setting on Arabic MMLU and EXAMs. As illustrated in Table 19, Turbo consistently outperforms other models in both MMLU and EXAMs benchmarks. Notably, Jais-13B-*chat* showcases the superior performance, which is consistent with the results in Sengupta et al. (2023). Specifically, its MMLU score stands at 37.11, trailing ChatGPT's score of 46.07 by a mere 8.96 points. On the EXAMs benchmark, Jais-13B-*chat* scored only 4.79 points lower than Turbo. One possible reason for Jais's good performance may be attributed to traditional NLP task datasets in their SFT dataset such as Super-NaturalInstructions Wang et al. (2022), which contains multiple-choice questions akin to the MMLU and EXAMs. Our model, in contrast, hasn't been trained on such data.

Table 18: Average F1 on **ACVA** in the few shot setting. The best performance is in **bold** and the second best is underlined.

| Model | All set | Clean set |
|---|---|---|
| Bloomz Muennighoff et al. (2022) | 58.94% | 60.91% |
| Jais-13B-base Sengupta et al. (2023) | 73.96% | 75.80% |
| **AceGPT-7B-base** | 74.72% | 70.32% |
| **AceGPT-13B-base** | 78.62% | 75.61% |
| Turbo | **80.12%** | **81.99%** |

Table 19: Accuracy of chat models on **Arabic MMLU** and **EXAMs**. The best is in **bold** and the second is underlined.

| Model | Arabic MMLU | | | | | EXAMs |
|---|---|---|---|---|---|---|
| | Average | STEM | Humanities | Social Sciences | Others (Business, Health, Misc) | |
| Phoenix | 27.52 | 23.92 | 25.56 | 32.15 | 28.47 | 31.6 |
| Phoenix-multiple-langs | 14.86 | 15.63 | 8.1 | 19.21 | 16.51 | 16.48 |
| Jais-13B-*chat* | 37.11 | 35.58 | 30.83 | 46.61 | 35.44 | 40.84 |
| AceGPT-7B-*chat* | 29.77 | 25.45 | 28.56 | 35.99 | 29.09 | 29.73 |
| AceGPT-13B-*chat* | 33.86 | 31.52 | 28.61 | 44.87 | 30.45 | 40.35 |
| Turbo | **46.07** | **44.17** | **35.33** | **61.26** | **43.52** | **45.63** |

### F.2 EVALUATION ON ARABIC NLU TASKS

**ALUE**    ALUE [9] is a popular online benchmark, which is similar to the GLUE benchmark but has a main focus on **A**rabic **L**anguage **U**nderstanding **E**valuation. It includes traditional NLP tasks such as sentiment analysis, semantic matching, text relation classification, and dialect identification. It comprises 9 tasks as illustrated in Table 20.

**Experiment setting**    We train our AceGPT-13B-*base* on each task independently in a fully supervised manner, resembling the approach of the top models on the leaderboard. Moreover, high-ranking models on the leaderboard adopt the grid search method on validation sets to select hyper-parameters. Similarly, we employ a Bayesian approach for hyperparameter adjustment. For tasks providing predefined validation split, we utilize the given validation sets. Otherwise, we allocate 10% of the data from the training set for validation purposes. For the DIAG task, which does not provide training data, we use the model trained on XNLI to evaluate on it.

**Experiment results and analysis**    Table 21 presents our performance on the ALUE benchmark. AceGPT ranks second in terms of the average score in these nine datasets, right behind AraMUS (Alghamdi et al. (2023)), which has conducted extensive pre-training in Arabic data. In future endeavors, we plan to incorporate a richer set of Arabic pre-training corpora and supervised data to enhance the model's NLU capabilities.

### G DETAILED RESULTS ON HUMAN EVALUATION

The results of the human evaluation corresponding to Table 8 for each annotator are shown in Table 22.

---

[9]https://www.alue.org/home

Table 20: Summary of NLU Tasks and Metrics in ALUE benchmark

| Task | Metric | Test Set Size | Ground Truth |
|------|--------|---------------|--------------|
| MQ2Q (NSURL-2019 Shared Task 8) | F1-score | 4000 | private |
| OOLD (OSACT4 Shared Task-A) | F1-score | 1000 | private |
| OHSD (OSACT4 Shared Task-B) | F1-score | 1000 | private |
| SVREG (SemEval-2018 Task 1) | Pearson correlation | 1000 | private |
| SEC (SemEval-2018 Task 1) | Jaccard similarity score | 1000 | private |
| FID (IDAT@FIRE2019) | F1-score | 1006 | public |
| MDD (MADAR Shared Task Subtask 1) | F1-score | 5200 | public |
| XNLI (Cross-lingual Sentence Representations) | Accuracy | 2490 | public |
| DIAG (Diagnostic dataset) | Matthews correlation | 1147 | public |

Table 21: Experimental results in ALUE Seelawi et al. (2021) including online baselines. While the leaderboard calculates the 'scores' excluding Task DIAG, we also incorporate it to derive the 'Avg'.

| Model | #Params | Avg | Score | MQ2Q | MDD | SVREG | SEC | FID | OOLD | XNLI | OHSD | DIAG |
|-------|---------|-----|-------|------|-----|-------|-----|-----|------|------|------|------|
| ARABIC-BERT | 135M | 63.5 | 67.1 | 85.7 | 59.7 | 55.1 | 25.1 | 82.2 | 89.5 | 61.0 | 78.7 | 19.6 |
| ARABERTv0.1-base | 135M | 64.2 | 68.4 | 89.2 | 58.9 | 56.3 | 24.5 | 85.5 | 88.9 | 67.4 | 76.8 | 23.5 |
| ARABIC-BERT | 110M | 68.6 | 69.3 | 89.7 | 59.7 | 58.0 | 26.5 | 84.3 | 89.1 | 67.0 | 80.1 | 19.0 |
| CAMeLBERT-MIX | 108M | 66.7 | 70.4 | 89.4 | 61.3 | 69.5 | 30.3 | 85.5 | 90.3 | 56.1 | 80.6 | 11.8 |
| AraT5-base | 289M | 67.6 | 71.1 | 91.3 | 63.8 | 65.9 | 30.5 | 82.3 | 88.8 | 68.2 | 77.9 | 15.4 |
| ARBERT | 163M | 65.5 | 71.4 | 89.3 | 61.2 | 66.8 | 30.3 | 85.4 | 89.5 | 70.7 | 78.2 | 24.3 |
| MARBERT | 163M | 63.9 | 72.2 | 83.3 | 61.9 | 75.9 | 36.0 | 85.3 | 92.1 | 64.3 | 78.9 | 12.3 |
| JABER | 135M | 68.2 | 73.7 | 93.1 | 64.1 | 70.9 | 31.7 | 85.3 | 91.4 | 73.4 | 79.6 | 24.4 |
| Char-JABER | 136M | 70.1 | 75.3 | 92.0 | 66.1 | 74.5 | 34.7 | 86.0 | 92.3 | 73.1 | 83.5 | 26.7 |
| ALM-1.0 | 350M | 70.3 | 75.8 | 94.5 | 65.1 | 70.1 | 35.3 | 86.0 | 91.7 | 77.7 | 85.7 | 30.2 |
| SABER | 369M | 71.4 | 77.3 | 93.3 | 66.5 | 79.2 | 38.8 | 86.5 | 93.4 | 76.3 | 84.1 | 26.2 |
| AraMUS | 11B | **74.0** | **79.8** | **95.2** | 67.5 | **80.4** | **41.6** | **87.2** | 95.5 | 83.2 | 87.4 | 42.0 |
| **AceGPT-13B-*base*** | 13B | 72.8 | 76.6 | 94.9 | 63.3 | 72.4 | 36.8 | 85.1 | 94.2 | 81.0 | 85.4 | **42.2** |

Table 22: Details of human evaluations on Arabic Vicuna-80 and Arabic AlpacaEval.

| Dataset | Comparison | | | win | tie | lose |
|---|---|---|---|---|---|---|
| Arabic Vicuna-80 | AceGPT-7B-chat vs. Jais-13B-chat | | volunteer 1 | 66 | 3 | 11 |
| | | | volunteer 2 | 65 | 9 | 6 |
| | | | volunteer 3 | 67 | 4 | 9 |
| | AceGPT-7B-chat vs. Turbo | | volunteer 1 | 26 | 0 | 54 |
| | | | volunteer 2 | 40 | 0 | 40 |
| | | | volunteer 3 | 0 | 79 | 1 |
| | AceGPT-7B-chat (w/o RLAIF) vs. Turbo | | volunteer 1 | 23 | 12 | 45 |
| | | | volunteer 2 | 12 | 58 | 10 |
| | | | volunteer 3 | 31 | 0 | 49 |
| | AceGPT-13B-chat vs. Jais-13B-chat | | volunteer 1 | 68 | 6 | 6 |
| | | | volunteer 2 | 65 | 5 | 10 |
| | | | volunteer 3 | 66 | 5 | 9 |
| | AceGPT-13B-chat vs. Turbo | | volunteer 1 | 14 | 35 | 31 |
| | | | volunteer 2 | 21 | 28 | 31 |
| | | | volunteer 3 | 4 | 74 | 2 |
| | AceGPT-13B-chat (w/o RLAIF) vs. Turbo | | volunteer 1 | 19 | 14 | 47 |
| | | | volunteer 2 | 22 | 19 | 39 |
| | | | volunteer 3 | 6 | 57 | 17 |
| Arabic AlpacaEval | AceGPT-7B-chat vs. Jais-13B-chat | | volunteer 1 | 515 | 196 | 94 |
| | | | volunteer 2 | 619 | 54 | 132 |
| | | | volunteer 3 | 146 | 632 | 27 |
| | AceGPT-7B-chat vs. Turbo | | volunteer 1 | 259 | 291 | 255 |
| | | | volunteer 2 | 71 | 632 | 102 |
| | | | volunteer 3 | 158 | 200 | 447 |
| | AceGPT-13B-chat vs. Jais-13B-chat | | volunteer 1 | 283 | 504 | 18 |
| | | | volunteer 2 | 400 | 343 | 62 |
| | | | volunteer 3 | 509 | 187 | 109 |
| | AceGPT-13B-chat vs. Turbo | | volunteer 1 | 216 | 326 | 263 |
| | | | volunteer 2 | 47 | 664 | 94 |
| | | | volunteer 3 | 346 | 84 | 375 |

