# OpenReview forum: "AceGPT, Localizing Large Language Models in Arabic"
_ICLR.cc/2024/Conference — ICLR 2024 Conference Withdrawn Submission_

### Official Review · Reviewer_vrrL · 2023-10-15

**Soundness:** 3 good
**Presentation:** 3 good
**Contribution:** 2 fair
**Rating:** 5
**Confidence:** 3

**Summary:**

The authors mainly did continue training/SFT/RLAIF to LLM to further improve LLM in Arabic. From my perspective, the main issue the authors want to solve is cultural difference. Training on specific language sources helps solve the cultural difference.

**Strengths:**

The authors organize the paper clearly and describe in details on how to tune an LLM to work better on a specific language Arabic. From the evaluation, tuning improves metrics on both human and auto evaluations.

**Weaknesses:**

I think the main weakness is novelty. Though novel application should be also considered an contribution, I do not think the paper provides many insights on LLM in Arabic. It seems to just follow the common techniques continue training/SFT/RLAIF.

**Questions:**

What insight can I get from the paper? Does the paper just follow continue training/SFT/RLAIF to make LLM work on Arabic?

---

### Official Review · Reviewer_P1tY · 2023-11-02

**Soundness:** 3 good
**Presentation:** 3 good
**Contribution:** 2 fair
**Rating:** 3
**Confidence:** 4

**Summary:**

Current LLMs have shown good performance on English but fail to align with local values and cultural norms in non-English environments. This paper introduces a packaged solution to localize LLMs to the Arabic language. First, they conduct an incremental pre-training on Arabic data. Then, they fine-tune Arabic natural questions and instructions related to Arab. Next, they employ a reward model trained on data with local culture and values.

The authors evaluate their model on different benchmarks, i.e., instruction-following benchmark, NLU benchmark, knowledge benchmark and culture localization benchmark. The model achieves SOTA performance among open Arabic LLMs.

**Strengths:**

1.	This paper is well-written and easy to follow.
2.	Localization on specific culture is an important topic in LLMs.
3.	This paper mentioned Arabic-related dataset and models, which can be useful for the people in related fields.

**Weaknesses:**

1.	The theoretical and technical contributions are poor. This paper is more like a engineering report to introduce how to localize a public LLM on Arabic, illustrating the operation and dataset during pre-training, instruction tunning and RLHF stage. All the methods are well-known. The findings are intuitive, using localize data to pre-train, instruction tuning and training RLHF can be helpful for better localization. It seems more suitable for an empirical NLP conference rather than a learning conference.
2.	The newly introduced Arabic Cultural and Value Alignment benchmark is not novel. Previous work [1] also introduce a benchmark to measure the culture bias in Arabic LLMs. It seems the authors didn’t mention and cite this paper.
3.	The analyses are insufficient. I will be more excited to see a comprehensive and systematic evaluation on the contribution of different operation, i.e., pre-training, instruction tuning and RLHF, on different evaluation perspective, such as instruction-following, knowledge accuracy, culture alignment, etc.

[1] Having Beer after Prayer? Measuring Cultural Bias in Large Language Models, https://arxiv.org/abs/2305.14456

**Questions:**

1.	Can you clarify your technical or theoretical contribution? Any novel method or new understanding about the previous method?
2.	Can you provide a comprehensive and systematic evaluation on the contribution of different operation, i.e., pre-training, instruction tuning and RLHF, on different evaluation perspective, such as instruction-following, knowledge accuracy, culture alignment, etc?

---

### Official Review · Reviewer_CfmQ · 2023-11-02

**Soundness:** 3 good
**Presentation:** 3 good
**Contribution:** 2 fair
**Rating:** 3
**Confidence:** 4

**Summary:**

This paper describes the building and evaluation of an LLM finetuned for the Arabic language and Arabic culture (referred to as localization). The work starts with a strong Llama2 model and does continued pre-training on a modest amount of English and Arabic data, supervised finetuning and RLAIF to obtain Arabic LLMs with improved instruction following capabilities, factual understanding and Arabic cultural sensitivity. AceGPT shows significant improvement over Jais (another Arabic), and is competitive with ChatGPT. The paper contributes a new benchmark ACVA for studying cultural alignment of language models. Localization/cultural alignment is a strong theme in the paper.

**Strengths:**

* Pre-training with Arabic data seems to improve performance on MMLU and ACVA benchmarks, proving the utility of native language data.
* An interesting analysis of preference for certain cultural contexts in Language model responses. A dataset for studying cultural alignment the same has been created, which is a novel and useful contribution.
* Better performance than JAIS, comparable to ChaptGPT.
* Ablation studies show the utility of pre-training with Arabic data and RLAIF (which improves both instruction following and localization)

**Weaknesses:**

* While the paper makes an interesting contribution to an improved Arabic LLM, it does little to advance the study of building/adapting LLMs for non-English languages. Most of the methods are well known. A few studies can help draw broader lessons on the localization of LLMs:
    * How much pre-training data is required? What is the best data balance between English and other languages?
    * Does the English performance get impacted due to the finetuning?
    * How objective is the ACVA benchmark? Different annotators (even amongst Arabic annotators) themselves might have different viewpoints on Arabic culture. On  subjective questions, what is the inter-annotator agreement on the answers?

* Evaluation using automatically translated datasets can be problematic (MMLU and EXAM).
  * Some human judgment on translation quality should be provided to understand the quality of the benchmark.
  * You could consider other translation models: GPT4 vs NLLB or any available Arabic-specific NMT model that might be better than GPT4.

**Questions:**

* Please add more details about JAIS, which is the competing model, and should be included for readability. A discussion on why ACEGPT is better. Jais is trained with much more Arabic data, but a lot less English data (compared to the backbone Llama model that ACEGPT uses). yes, AceGPT has better performance than JAIS – which takes the approach of training from scratch and uses far less English data than  Llama2.
* ACVA dataset: The dataset and its construction can be described better to better understand them.
     * You could put some examples of the ACVA tes test set (questions as well as responses (both yes and no) (there is a single example in E.4, but in Arabic). Multiple examples (both yes/no) with English translations would be very useful.
     * You could add some examples of the prompt for ACVA dataset generation.
* For all tasks, can you add the base/SFT/RLAIF model results? Would be useful for comparison and understanding impact.
* Section 4.3.1: There seems to be little difference in the reward between ‘yes’ and ‘no’ answers – yet the model seems to improve on Arabic cultural alignment. Is it just due to the fact that the last stage of finetuning consumed Arabic Quora data? If an additional SFT training was done on just the Arabic Quora data, would that have had the same impact as RLAIF?